# Comparison of Tumour-Specific Phenotypes in Human Primary and Expandable Pancreatic Cancer Cell Lines

**DOI:** 10.3390/ijms241713530

**Published:** 2023-08-31

**Authors:** Feng Guo, Kejia Kan, Felix Rückert, Wolfgang Rückert, Lin Li, Johannes Eberhard, Tobias May, Carsten Sticht, Wilhelm G. Dirks, Christoph Reißfelder, Prama Pallavi, Michael Keese

**Affiliations:** 1Department of Surgery, Universitätsmedizin Mannheim, Medical Faculty Mannheim, Heidelberg University, 68167 Mannheim, Germany; feng.guo@medma.uni-heidelberg.de (F.G.); kejia.kan@medma.uni-heidelberg.de (K.K.); lin.li@medma.uni-heidelberg.de (L.L.); johannes.eberhard@umm.de (J.E.); christoph.reissfelder@umm.de (C.R.); 2European Center of Angioscience ECAS, Medical Faculty Mannheim, Heidelberg University, 68167 Mannheim, Germany; 3Surgical Department, Diakonissen Krankenhaus Speyer, 67346 Speyer, Germany; felix.rueckert@diakonissen.de; 4Ingenieurbüro Dr. Ing. Rückert Data Analysis, Kirchweg 4, 57647 Nistertal, Germany; info@ib-rueckert.de; 5InSCREENeX GmbH, Inhoffenstr. 7, 38124 Braunschweig, Germany; tobias.may@inscreenex.com; 6Next Generation Sequencing Core Facility, Medical Faculty Mannheim, Heidelberg University, 68167 Mannheim, Germany; carsten.sticht@medma.uni-heidelberg.de; 7Leibniz Institute DSMZ, German Collection of Microorganisms and Cell Cultures GmbH, Inhoffenstraße 7B, 38124 Braunschweig, Germany; wdi@dsmz.de; 8Department of Vascular Surgery, Theresienkrankenhaus, 68165 Mannheim, Germany

**Keywords:** chemosensitivity, RNA-Seq, patient-tailored therapy, chemotherapy, regression analysis, statistical comparison of populations

## Abstract

There is an ongoing need for patient-specific chemotherapy for pancreatic cancer. Tumour cells isolated from human tissues can be used to predict patients’ response to chemotherapy. However, the isolation and maintenance of pancreatic cancer cells is challenging because these cells become highly vulnerable after losing the tumour microenvironment. Therefore, we investigated whether the cells retained their original characteristics after lentiviral transfection and expansion. Three human primary pancreatic cancer cell lines were lentivirally transduced to create expandable (Ex) cells which were then compared with primary (Pri) cells. No obvious differences in the morphology or epithelial–mesenchymal transition (EMT) were observed between the primary and expandable cell lines. The two expandable cell lines showed higher proliferation rates in the 2D and 3D models. All three expandable cell lines showed attenuated migratory ability. Differences in gene expression between primary and expandable cell lines were then compared using RNA-Seq data. Potential target drugs were predicted by differentially expressed genes (DEGs), and differentially expressed pathways (DEPs) related to tumour-specific characteristics such as proliferation, migration, EMT, drug resistance, and reactive oxygen species (ROS) were investigated using the Kyoto Encyclopedia of Genes and Genomes (KEGG) database. We found that the two expandable cell lines expressed similar chemosensitivity and redox-regulatory capability to gemcitabine and oxaliplatin in the 2D model as compared to their counterparts. In conclusion, we successfully generated expandable primary pancreatic cancer cell lines using lentiviral transduction. These expandable cells not only retain some tumour-specific biological traits of primary cells but also show an ongoing proliferative capacity, thereby yielding sufficient material for drug response assays, which may provide a patient-specific platform for chemotherapy drug screening.

## 1. Introduction

Recent studies have shown that pancreatic tumour cells can be singularised and cultured from tumour specimens. These may be used to predict the response towards systemic treatment in clinical settings [1]. To allow their use in therapy response prediction assays, it is important that cells retain and express most of the differentiated properties typical of their original source [2,3]. The culture efficacy of primary tumour cell lines depends on the availability of surgical materials [4], and the yields of cell cultivation are normally low depending on the isolation techniques and tumour biology. In pancreatic cancer, some primary cell lines lose their original traits over time [5]. In particular, pancreatic ductal adenocarcinoma cells are highly sensitive and lack robustness against changes in the tumour microenvironment after isolation [6]. In addition, these primary cells are divided only a limited number of times [7], which further complicates their use.

To yield sufficient material for the experiment, the immortalisation of primary cells seems to be an alternative. The transformation of a primary cell into an immortalised cell can be induced by a second oncogene [8], or, at a low frequency, by chromosomal mutations [9]. Although immortalised cells grow faster than primary cells [10], traditional approaches for establishing immortalised cell lines usually require genome manipulation, which results in changes in essential biological and genetic characteristics [11]. Compared with immortalised cells, expandable cells, as defined by the introduction of intercalating targeted genes into primary cells to gain robustness, can provide sufficient material and retain the majority of their original cell biological characteristics. Currently, we have established a method using a small lentiviral gene library to expand primary cells derived from different tissues, donors, and species [12]. In only 6 weeks, personalised cell lines can be generated from only 1×10^6^ primary cells. Therefore, this novel approach not only allows for the reproducible expansion of primary cells but also overcomes the unpredictability that is typically correlated with previous cell line development procedures [12].

Considering that expandable cells have also undergone some genomic mutations, it remains to be shown whether these cells can be used for response prediction in terms of planning individualised therapy as their genome is different from that of primary cells [13,14]. Therefore, in our study, expandable cell lines were created from human primary pancreatic cancer cell lines via lentiviral transduction using a small gene library. We compared the differences in primary and expandable cell lines using RNA-Seq data and tumour-specific phenotypes, such as morphology, proliferation, migration, epithelial–mesenchymal transition (EMT), chemotherapeutic response, and redox-regulatory capability.

## 2. Results

### 2.1. Growth Characteristics of Primary and Expandable Pancreatic Cancer Cell Lines

First of all, Figure 1 shows the flowchart of this study. The cell morphology was then determined using a 2D culture (Figure 2A). Primary MaPac107 (Pri-MaPac107) exhibited epithelial monolayer characteristics with a fusiform-organised pattern. Cells grew as clusters before reaching 100% confluency. Another characteristic was the presence of irregularly shaped nuclei. A fusiform-organised pattern did not appear when the cells reached confluence. Expandable MaPac107 (Ex-MaPac107) cells shared the same cell and nuclear morphology. Primary PaCaDD159 (Pri-PaCaDD159) cells proliferated as an epithelial monolayer with a lumpy organised pattern and grew in the form of a cluster-like unit. The cells were ovoid and were characterised by the presence of small and round cell nuclei. Expandable PaCaDD159 (Ex-PaCaDD159) shares the same cell and nuclear morphology. Primary PaCaDD165 (Pri-PaCaDD165) cells were grown in an epithelial monolayer as small polygonal cells with a cobblestone pattern. These cells exhibited prominent nuclei and rapid proliferation. No morphological differences were detected between the expandable PaCaDD165 (Ex-PaCaDD165) and Pri-PaCaDD165. Figure 2B,C show the growth curves and doubling times of all the cell lines. Ex-MaPac107 and Ex-PaCaDD165 expressed higher proliferation rates than their counterparts. The proliferation rate of Ex-PaCaDD159 cells was significantly lower than that of Pri-PaCaDD159 cells. Although the lower proliferation rate of Ex-PaCaDD159 cells was unexpectable, the remaining two expandable cell lines showed higher proliferation rates in 2D culture.

The cells were further evaluated in 3D cultures. Figure 3A shows that both Pri-MaPac107 and Ex-MaPac107 formed compact spherical structures on day 1. This structure became tighter over time, resulting in a smooth morphology on day three. Moreover, we observed the formation of a dark core in the centre of the spheroid on day 5, which gradually expanded until day 7. Although the diameters of Pri-MaPac107 and Ex-MaPac107 spheroids increased over time, the diameter of Ex-MaPac107 spheroids was larger than that of Pri-MaPac107 spheroids during the observation period (Figure 3B). Pri-PaCaDD159 did not form spheroids but formed irregularly shaped cell aggregates. Ex-PaCaDD159 did not form spheroids until day seven. Pri-PaCaDD165 formed a relatively loose spherical structure on day 1, subsequently becoming tighter and shrinking in diameter. A smooth spheroid was formed on day 3. Ex-PaCaDD165 formed loose spherical structures on day 1 and spheroids on day 3. Dissociation of the outer layer of the Ex-PaCaDD165 spheroids was observed on day 7. The size of the spheroids decreased on day 2 and gradually increased until day 7 (Figure 3B). In summary, neither Pri-PaCaDD159 nor Ex-PaCaDD159 formed spheroids in this study. The diameters of Ex-MaPac107 and Ex-PaCaDD165 spheroids were larger than those of their primary counterparts from day 1 to day 7.

To compare the migration abilities of primary and expandable pancreatic cancer cell lines, scratch assays were performed. Time-lapse images were taken at five time points (Figure 4A). Gap closure was faster in all three primary cell lines than in their expandable counterparts from 3 h to 9 h. At 24 h, all gaps were fully closed, except for those in Pri-PaCaDD165 and Ex-PaCaDD165 (Figure 4B).

In addition to cell migration, the invasiveness of a cell line determines its aggressiveness in cancer progression. EMT is a hallmark of tumour cell invasion. The expression levels of EMT-related genes of primary and expandable cells were determined by qPCR. Figure 5 depicts the expression levels of different EMT-related genes with GAPDH as an internal reference.

### 2.2. Bioinformatics Analysis Based on RNA-Seq Data

To determine differences in gene expression between primary and expandable pancreatic cancer cell lines, we performed bioinformatics analysis of RNA-Seq data. Hierarchical clustering using heatmaps depicted the primary and expandable samples as distinct clusters (Appendix A). In total, 2742, 649, and 1952 significantly DEGs were identified in MaPac107, PaCaDD159, and PaCaDD165, respectively. These were used to generate volcano plots (Appendix A) and Venn diagrams (Appendix A). Three-dimensional principal component analysis (PCA) showed the distribution pattern of the primary and expandable cell lines (Appendix A). Several differentially expressed pathways (DEPs) related to tumour-specific phenotypes were observed in a cell-line-specific manner (Table 1). The Hippo signalling pathway, related to drug resistance and proliferation, was enriched in MaPac107. Focal adhesion was enriched in MaPac107, influencing pathways related to EMT and migration. The NF-kappa B signalling pathway was enriched in PaCaDD159, which is related to drug resistance, EMT, proliferation, reactive oxygen species (ROS), and migration. In addition, the PI3K-Akt signalling pathway, enriched in PaCaDD159, was correlated with drug resistance, EMT, proliferation, ROS, and migration. The AMPK signalling pathway was enriched in PaCaDD165, which affects drug resistance and ROS production. The adjusted *p*-values for the aforementioned pathways were all <0.05.

### 2.3. Chemosensitivity of Primary and Expandable Pancreatic Cancer Cell Lines

Based on the DEGs derived from RNA-Seq data with logFC > 3 or <−3 and adjusted *p*-values < 0.05, we predicted potential target drugs using DGIdb. This analysis showed that 94 potential target drugs were shared between MaPac107, PaCaDD159, and PaCaDD165 (Figure 6A). Fourteen compounds are being clinically used (Appendix A), and gemcitabine and oxaliplatin are administered in adjuvant or palliative settings in pancreatic cancer.

Therefore, we focused on the effects of gemcitabine and oxaliplatin in this study. For both compounds, different concentrations (Appendix A) were chosen to perform IC_50_ assays. The dose–response curves of primary and expandable cells aligned for MaPac107 and PaCaDD165 after 48 and 72 h of incubation with gemcitabine and oxaliplatin (Figure 6B,D). The dose–response curves for Pri-PaCaDD159 and Ex-PaCaDD159 were not aligned at these two time points after exposure to either compound (Figure 6C). The IC_50_ values are summarised in Appendix A. The DEGs related to gemcitabine and oxaliplatin are listed in Appendix A. Both primary and expandable cell lines display identical drug response behaviour. The detailed comparison of cell line response to drugs - gemcitabine and oxaliplatin and statistical calculations including hypothesis definition, group description Appendix A, variance Appendix A, outlier percent calculation Appendix A, example of various regression methods employed Appendix A, and the descriptive parameters for the regression methods Appendix A and finally an example for results of the t-test for residuals Appendix A are presented in the Appendix A under section cell line comparison.

### 2.4. Cellular Redox Status Assessment

To assess the cellular redox status of primary and expandable cells under oxidative stress, the cells were transfected with Grx1-roGFP3. The fluorescence intensity was determined by calculating the ratio of E_GSH_ = E_x_395/E_x_485. We initially monitored the dynamics of E_GSH_ using H_2_O_2_ and DTT to validate the function of this sensor after sorting (Appendix A). We then compared the redox-regulatory capability of primary and expandable cells after chemotherapy in 2D culture. Significant differences in E_GSH_ activity were found between Pri-MaPac107-roGFP3+ and Ex-MaPac107-roGFP3+ cells after 48 h of treatment with gemcitabine and oxaliplatin (Figure 7A,B). Figure 7C,D show the responses of Pri-PaCaDD159-roGFP3+ and Ex-PaCaDD159-roGFP3+ cells to gemcitabine and oxaliplatin after 48 h of treatment, which showed a similar fluorescence intensity coefficient for E_GSH_. Figure 7E,F show that E_GSH_ for Pri-PaCaDD165-roGFP3+ and Ex-PaCaDD165-roGFP3+ after 48 h of treatment showed differences after exposure to gemcitabine, while exposure to oxaliplatin led to similar E_GSH_.

Subsequently, we compared the redox-regulatory capability of primary and expandable cells in 3D culture after incubation with gemcitabine and oxaliplatin for 48 h. Figure 8A,B show obvious differences between Pri-MaPac107-roGFP3+ and Ex-MaPac107-roGFP3+. A disaggregated outer layer of spheroids of Pri-MaPac107-roGFP3+ cells was observed after incubation with gemcitabine, which was not observed in Ex-MaPac107-roGFP3+ cells (Appendix A). Figure 8C,D depict clear differences between Pri-PaCaDD165-roGFP3+ cells and Ex-PaCaDD165-roGFP3+ cells. No obvious disaggregation on the outer layer of spheroids was observed in both treatment groups of Pri-PaCaDD165-roGFP3+ and Ex-PaCaDD165-roGFP3+ (Appendix A).

## 3. Discussion

Primary cells derived from tumour specimens have been widely used for studies of drug metabolism and toxicity in vitro because they endogenously express drug targets of interest at levels consistent with in vivo conditions [3]. One of the drawbacks is that it usually takes a long time for primary tumour cells to expand [63]. Moreover, some cell lines become static or senescent after a certain number of passages [6]. Although immortalised cells can be an option for infinite proliferation, they generally require highly expressed viral oncogenes and lead to alterations in the cellular phenotype and chromosomal instability [64,65]. To overcome these shortcomings, a new method called targeted transgenic transfection may be an alternative. In our study, we used a lentiviral library consisting of 33 genes to establish three expandable primary pancreatic cancer cell lines and expected them to maintain the prominent biological characteristics of their primary counterparts. This method is widely used to expand human primary chondrocytes, epithelial cells, endothelial cells, and murine primary hepatocytes. Expansion targets diverse cellular processes, such as cell cycle progression and apoptosis, but allows cells to maintain stem cell properties and overcome the problem of chromosomal instability [12]. No signs of senescence or crisis, tumour formation, or pluripotent phenotype were observed in the investigated cell lines during the extended cultivation periods. Expansion enables primary tumour cells to survive and proliferate outside the cellular tumour microenvironment. The advantage is the ongoing proliferative capacity and, thus, the availability of sufficient material for therapy. The question is whether these expandable cells can be used as reliable platforms for therapy response prediction. To evaluate the differences between primary and expandable cells, we compared three pancreatic carcinoma primary and expandable cell lines in terms of morphology, proliferation, migration, EMT, RNA-Seq, susceptibility towards chemotherapy reagents, and redox status assessment.

Although the morphologies of all expandable cell lines shared a similar pattern with their counterparts, we showed that Ex-MaPac107 and Ex-PaCaDD165 proliferated faster than their counterparts. Unexpectedly, Ex-PaCaDD159 exhibited slower proliferation and a longer doubling time than Pri-PaCaDD159. For optimal growth, cells in a solid tumour are dependent on the tumour microenvironment and interact with other cell types, such as fibroblasts, immune cells (T and B lymphocytes, natural killer cells, and tumour-associated macrophages), blood vessels, extracellular matrix (ECM), and other signal molecules [66,67]. Therefore, the relatively low proliferation rate of Pri-PaCaDD159 may be explained by its extraction history, that is, from a solid tumour, as compared with Pri-MaPac107 and Pri-PaCaDD165 which originate from pleural effusion and ascites. Subsequently, we investigated whether the primary and expandable cells could form 3D spheroids. In our study, both primary and expandable MaPac107 and PaCaDD165 formed homotypic tumour spheroids (single-cell type). Both Pri-PaCaDD159 and Ex-PaCaDD159 failed to form spheroids. For spheroid formation, tumour cells interconnect with each other through the formation of desmosomes and dermal junctions [68], as well as the secretion and deposition of proteoglycans and ECM proteins such as collagen, fibronectin, tenascin, and laminin [69]. The growth of spheroids normally shows an initial phase of volume increase, followed by a period known as the ‘spheroidization/stabilization time’ [70]. During spheroidisation, spheroids transformed into a more regular shape, which was observed in primary and expandable MaPac107 and PaCaDD165. Moreover, spheroids have a well-defined spatial structure that encompasses an actively proliferative outer layer as a result of the high availability of oxygen and nutrients, a middle layer consisting of quiescent and senescent cells, and an inner apoptotic/necrotic core due to the restricted distribution of nutrients and oxygen [71,72]. The proliferative outer layer and the apoptotic/necrotic core were clearly observed in the spheroids of Ex-PaCaDD165, Pri-MaPac107 and Ex-MaPac107 on day 7, respectively.

Thereafter, differences in cell migration and EMT between primary and expandable cells were examined. The migration of tumour cells is a prerequisite for tumour metastasis. Cell protrusions, chemotaxis, and cell polarity are significant molecular bases for migration in 2D tumour cell cultures [73]. We found that the migratory ability of the three expandable cell lines was attenuated in 2D culture compared to that of the primary cell lines. The lentiviral transduction technology used in this study may affect the biological function of expandable cell lines from the gene level. Apart from this, some DEPs may play a critical role in the attenuated migratory ability, which was listed below. EMT is a biological process in which non-motile polarised epithelial cells undergo a series of biochemical alterations, turning into motile non-polarised mesenchymal cells with invasive capability, resistance to apoptosis, and adjusted biosynthesis of ECM components [74]. SNAI2, TWIST1, and ZEB1 are the key regulatory genes contributing to the EMT process [75], where SNAI2 silencing substantially inhibits the EMT process [76], and overexpression of constitutively active TWIST1 in tumour cells promotes the acquisition of conspicuous EMT characteristics [77]. The overexpression of ZEB1 results in the enhanced migration, invasion, and EMT of pancreatic cancer cells [78]. While TJP1 expression in tumour cells is repressed in activated EMT [79], we found that the expression levels of SNAI2 and TWIST1 were significantly decreased in Ex-MaPac107 and the expression of ZEB1 was significantly increased in Ex-MaPac107 compared with Pri-MaPac107. The expression patterns of SNAI2, TWIST1, and ZEB1 in Ex-PaCaDD159 were comparable to those in Ex-MaPac107. As for PaCaDD165, we found that SNAI2 expression was significantly enhanced in Ex-PaCaDD165 and ZEB1 expression was significantly lower in Ex-PaCaDD165. The expression levels of TWIST1 in the primary and expandable cells were not significantly different. Therefore, EMT was not obviously affected by Ex-PaCaDD165. Interestingly, multiple DEPs in our KEGG database were related to proliferation, migration, and EMT, with adjusted *p*-values < 0.05. Examples include the Toll-like receptor [26,44,56], PI3K-Akt [27,35,54], Hedgehog [28,40,60], JAK-STAT [29,41,55], TGF-β [30,43,61], focal adhesion [31,58], Hippo [33,38,57], NF-kappa B [34,36,59], EGFR tyrosine kinase inhibitor resistance [37,53], MAPK [39], Ras [42], and chemokine [45,62] pathways found in the KEGG database of MaPac107. The Toll-like receptor [26,44,56], PI3K-Akt [27,35,54], JAK-STAT [29,41,55], focal adhesion [31,58], neutrophil extracellular trap formation [32], NF-kappa B [34,36,59], and chemokine [45,62] pathways were found in the KEGG database of PaCaDD159. All these pathways may play a critical role in the mediation of proliferation, migration, and EMT. No related pathways were identified in the KEGG database of PaCaDD165.

To predict drugs which might be therapeutically efficient, we analysed RNA-Seq data using the DGIdb. Fourteen potential target drugs related to cancer were identified. Gemcitabine and oxaliplatin were used to determine the chemosensitivity of the cells. While Pri-MaPac107 and Ex-MaPac107 and Pri-PaCaDD165 and Ex-PaCaDD165 shared similar chemosensitivities towards gemcitabine and oxaliplatin, differences in chemosensitivity were found between Pri-PaCaDD159 and Ex-PaCaDD159. Interestingly, two DEGs were identified which may help explain the difference in sensitivity towards gemcitabine and oxaliplatin in PaCaDD159. Tumour necrosis factor-alpha (TNF-α) expression correlates with sensitivity towards gemcitabine. Its expression by Ex-PaCaDD159 was up-regulated compared to Pri-PaCaDD159. TNF-α enhances the invasiveness of pancreatic cancer cells in vitro and promotes tumour growth and metastasis in mouse models of orthotopic pancreatic cancer [80]. It has been shown that the combination of etanercept (an inhibitor of TNF-α) and gemcitabine failed to enhance gemcitabine efficacy in advanced pancreatic cancer [81]. The combination of AdEgr.TNF.11D (adenoviral vector expressing human TNF-α) and gemcitabine enhanced antitumour activity in human pancreatic tumour models [82]. The IC_50_ values of Ex-PaCaDD159 for gemcitabine at 48 h and 72 h were higher than those of Pri-PaCaDD159, which correlates with a higher expression of TNF-α. In this context, gemcitabine treatment has been shown to increase TNF-α mRNA expression in tumour cells [83]. Another upregulated gene of interest in Ex-PaCaDD159 is CXCL10, which is related to sensitivity to oxaliplatin. High expression of CXCL10 mRNA in vivo correlates with higher sensitivity to oxaliplatin and capecitabine [84]. CXCL10 is an important angiostatic chemokine involved in tumour growth and new vessel formation. CXCL10-derived peptides do not only inhibit vessel formation and induce the involution of newly formed vessels [85]; these deformed and abnormal vessels may decrease the ingestion of chemotherapeutic drugs in tumour lesions too. In addition, some DEPs related to drug resistance may also help to explain the differences in the chemo-responsiveness of Pri-PaCaDD159 and Ex-PaCaDD159, including NF-kappa B [15], PI3K-Akt [17], necroptosis [20], nucleotide excision repair [21], and JAK-STAT [23], with adjusted *p*-values of less than 0.05.

Chemotherapy may cause oxidative stress, and high ROS levels may be detrimental to cancer cells. This is the dominant mechanism in many types of chemotherapeutics [86]. ROS are not only continually produced and removed in biological systems but are also required to drive certain regulatory pathways which are also related to cell survival [87]. We found that some DEPs in our KEGG database were also related to ROS production or ROS-induced apoptosis in pancreatic cancer cells, with an adjusted *p*-value of less than 0.05. The JAK-STAT [46], MAPK [47,48], PI3K/Akt [49], NF-kappa B [50], Lysosome [51], AMPK [52], and FOXO1 [25] pathways were found in the KEGG database of MaPac107. The JAK-STAT [46], PI3K/Akt [49], and NF-kappa B [50] pathways were identified in the KEGG database of PaCaDD159. The AMPK [52] pathway was identified using the KEGG database of PaCaDD165. A roGFP redox sensor was used to determine the influence of chemotherapeutics on the redox environment in primary and expandable cells. This sensor is a flexible platform for dynamic in situ measurements of cellular redox environment [88]. In particular, the Grx1-roGFP2 fusion protein has been reported to allow the dynamic live imaging of E_GSH_ to take place in different cellular compartments with high sensitivity and temporal resolution. The glutaredoxin1 (Grx1) confers dynamic responsiveness to the glutathione (GSH)/glutathione disulfide (GSSG) redox state [89]. E_GSH_ depends on the concentration of GSH and the ratio of GSH to GSSG [90], and an increased ratio of GSH/GSSG is indicative of greater oxidative stress. Furthermore, cells expressing Grx-roGFP3 showed an improved fluorescence signal intensity compared with cells expressing Grx-roGFP2 [91]. In our study, tumour cells were transfected with Grx1-roGFP3 and exposed to gemcitabine and oxaliplatin for 48 h to monitor the variation in E_GSH_ in 2D and 3D cultures. Significant differences were identified between Pri-MaPac107-roGFP3+ and Ex-MaPac107-roGFP3+ under exposure to the two drugs regardless of whether the cells were maintained in 2D or 3D culture. One explanation may be the differential expression of members of oxidative and antioxidative genes. We found that ABCG2 was upregulated in Ex-MaPac107 cells and was related to gemcitabine and oxaliplatin. It has been reported that cells expressing ABCG2 under oxidative stress elevate superoxide radical levels and further influence ROS production to the point of oxidant stress [92,93]. This might be a protective mechanism of Ex-MaPac107, which can protect cells from death caused by excessive ROS. A recent study has proven that ABCG2 is capable of protecting cells from ROS-mediated cell damage and death [94]. Interestingly, the fluorescence intensity of Ex-PaCaDD159 decreased after incubation with gemcitabine in 2D cultures. This decrease was correlated with increasing concentrations of gemcitabine. In this context, it has been shown that the treatment of tumour cells with Se-Gem, a selenoprodrug of gemcitabine, also triggers a dose-dependent decrease in the GSH/GSSG ratio [95], which is in line with our data. A possible explanation is that the ratio of GSH/GSSG decreases during cell death because of NADPH oxidation and GSH extrusion, and the exogenous addition of GSSG has also been shown to induce apoptosis [96,97]. While oxaliplatin is usually applied in combination with 5-FU, leucovorin and irinotecan for pancreatic cancer treatment (FOLFIRINOX) [98], the treatment with oxaliplatin alone still triggered oxidative stress in our study since the fluorescence intensity ratios in Pri-PaCaDD159-roGFP3+ and Ex-PaCaDD159-roGFP3+ and Pri-PaCaDD165-roGFP3+ and Ex-PaCaDD165-roGFP3+ were enhanced in correlation to the concentration of oxaliplatin. It has also been reported that an increased GSH/GSSG ratio was found in the human ovarian cancer cell line A2780 after combination treatment with LH3, a new monofunctional planaramine platinum (II) complex with curcumin [99]. This is also in accordance with our findings because oxaliplatin is also a platinum compound.

Interestingly, more differences in fluorescence ratios between Pri-PaCaDD165-roGFP3+ and Ex-PaCaDD165-roGFP3+ were observed in 3D culture compared to 2D culture after incubation with oxaliplatin for over 48 h. Generally, drugs or stimuli can readily reach cells in 2D culture, in which cells are exposed to drugs in a monolayer structure. In spheroids, the transport of nutrients, oxygen, and drugs depends on diffusion and concentration gradients, hydraulic conductivity, and pressure gradients [100]. The diameter of a spheroid is an essential factor which defines the extent to which a substance can reach cells within the spheroid [101]. Drugs may easily penetrate the outer layer of the spheroid and hardly penetrate the middle or deeper layer of the spheroid, which implies that the drugs cannot affect cells within the spheroids. The differences observed in our study between 2D and 3D cell cultures may also be attributed to the different assays used, since it is impossible for the detector of the SPARK Plate Reader to capture all the single-cell fluorescence from the spheroids.

## 4. Materials and Methods

### 4.1. Cell Culture

Human primary pancreatic cancer cell lines (MaPac107 and PaCaDD159. PaCaDD165) were originally derived from patient tumour tissues [102] (Table 2). The cells were cultured in Dresden medium [6] consisting of Dulbecco’s Modified Eagle Medium (DMEM, 4.5 g/L glucose, Sigma-Aldrich, Taufkirchen, Germany) and Keratinocyte serum-free medium (KSFM, Thermo Fisher Scientific, Waltham, MA, USA) at a ratio of 2:1. DMEM was supplemented with 1% penicillin/streptomycin (P/S; Sigma-Aldrich, Taufkirchen, Germany) and 20% foetal bovine serum (FBS; Thermo Fisher Scientific, Waltham, MA, USA). KSFM was supplemented with human recombinant epidermal growth factor (rEGF) and bovine pituitary extract (BPE). The cells were maintained in an atmosphere of 5% CO_2_ at 37 °C and passaged at a 1:3 ratio. The cell culture medium was changed every two days.

### 4.2. DNA Fingerprinting (STR Typing of Genomic DNA)

STR DNA profiling was carried out for the MaPac107 primary cell line, after establishment, using fluorescent PCR in combination with capillary electrophoresis, as described previously [103]. Using different alternate colours, the PowerPlex VR 1.2 system (Promega, Mannheim, Germany) was modified in order to run a two-colour DNA profiling, allowing the simultaneous single-tube amplification of eight polymorphic STR loci and Amelogenin for gender determination. The STR loci of CSF1PO, TPOX, TH01, vWA, and Amelogenin were amplified by primers labelled with the Beckman/Coulter dye D3 (green; Sigma-Aldrich, Taufkirchen, Germany), while the STR loci D16S539, D7S820, D13S317, and D5S818 were amplified using primers labelled with D2 (black). All the loci except the Amelogenin gene in this set are true tetranucleotide repeats. All primers are identical to the PowerPlexVR 1.2 system except the fluorescent colour. Data were analysed with the CEQ 8800 software (Beckman-Coulter, Krefeld, Germany), which enables an automatic assignment of genotypes and automatic export of resulting numeric allele codes into the reference DNA database of the DSMZ.

### 4.3. Expandable Cells

Expandable cells were produced using InSCREENeX via infection with a lentiviral library consisting of 33 genes [12]. Virus production was performed for each lentiviral vector individually by the transient transfection of HEK 293T cells using plasmids encoding helper functions (gagpol, rev, VSV-G) and the respective lentiviral vectors [104]. All the expandable cells were cultured in Dresden medium at 37 °C in a 5% CO_2_ atmosphere, being passaged at a 1:3 ratio.

### 4.4. Doubling Time

The doubling time was assessed by counting the number of viable cells from freshly trypsinised monolayers using a haemocytometer. Cell viability was determined by trypan blue staining (Sigma-Aldrich, Taufkirchen, Germany). A total of 50,000 cells from each cell line were seeded in 6-well plates with 2 mL of Dresden medium per well and counted every 24 h for 7 days. The culture medium was changed every three days. The doubling time was calculated from the logarithmic growth curve using the following formula [102]: v = logN − logN_0_/log2 (t − t_0_), with doubling time = 1/v, where N is the number of cells and t is the time. Each well of the plate was counted three times. Experiments were performed in triplicate.

### 4.5. Three-Dimensional Spheroid Establishment

In total, 3000 cells in 100 μL of Dresden medium containing 20% methylcellulose (Sigma-Aldrich, Taufkirchen, Germany) were seeded in 96-well U-bottom plates (Corning, NY, USA) and centrifuged at 2000× *g* for 15 min. The plates were incubated at 37 °C in a 5% CO_2_ atmosphere. Every two days, 50% Dresden medium was changed to maintain proliferation and viability in all plates.

### 4.6. Scratch Assay

A total of 100,000 cells were seeded in 12-well plates with 1 mL of Dresden medium per well. The cells were allowed to grow to approximately total confluence. A scratch was made by scraping the cell monolayer in a straight line using a 200 μL pipet tip. The debris and lost cells from the scratch margins were removed by washing the cells twice with 1 mL of Dresden medium. Images were taken by Carl Zeiss Axio Vert. A1 microscope (431030-9040-000; Jena, Germany) at 0, 3, 6, 9, and 24 h. Wound closure was determined as follows: wound closure (%) = (original wound area—area at each time point)/original wound area. Experiments were performed in triplicate.

### 4.7. RNA Isolation and Sequencing

RNA was isolated following the manufacturer’s instructions using a RNeasy Mini Kit (Qiagen, Hilden, Germany). Each cell line contained three RNA samples obtained from three sequential passages. RNA samples were quantified using nucleic acid quantification analysis. The purity was determined by measuring the absorbance ratio at 260 and 280 nm with acceptable values of 1.7–2.1 using a SPARK Plate Reader (Tecan V2.3, Männedorf, Switzerland). RNA integrity was assessed and analysed by capillary electrophoresis using an Agilent 2100 Bioanalyzer (Agilent Technologies, Santa Clara, CA, USA). An RNA Integrity Number (RIN) ≥7.0 indicated sufficient RNA quality. All samples were then sent to BGI Tech Solution Co., Ltd. (Hong Kong, China) for RNA-Seq.

### 4.8. Real-Time Quantitative PCR (qPCR)

RNA (500 ng) was used for cDNA synthesis using the Bio-Rad T100 (Hercules, CA, USA) following the manufacturer’s instructions using the QuantiTect Rev. Transcription Kit (Qiagen, Hilden, Germany). qPCR was performed using a LightCycler^®^ 96 (Roche, Basel, Switzerland) with the FastStart Essential DNA Green Master Kit (Roche, Basel, Switzerland) and primers (Qiagen, Hilden, Germany). Original data were obtained using the LightCycler^®^ 96 SW 1.1 (Roche, Basel, Switzerland). Relative mRNA expression levels were determined using the comparative Ct method after normalisation to GADPH expression levels. The expression levels of each gene were analysed in triplicate. The primers used in this study are listed in Table 3.

### 4.9. RNA-Seq Data Bioinformatics Analysis

In brief, the R statistical programming language and Bioconductor tools were used by employing the NGS analysis package system PipeR [105]. Quality control of the raw sequencing reads was performed using FastQC (version 0.11.5). Low-quality reads were removed using the Trim Galore (version 0.6.4). The resulting reads were aligned to the reference human genome (GRCh38.p13) from the gene code and counted using Kallisto (version 0.46.1) [106]. The count data were transformed to log2 counts per million (logCPM) using the voom function of the limma package [107]. Differential expression analysis was performed using the limma package in R software (version 3.6.3). A false-positive rate of α = 0.05 with FDR correction was taken as the level of significance. Volcano plots and heat maps were created using the ggplot2 package (version 2.2.1) and a complex heat map (version 2.0.0) [108]. Pathway analysis was performed using the fgsea package [109] and the enrichment browser package [110] in R software (version 3.6.3) using the pathway information from the Kyoto Encyclopedia of Genes and Genomes (KEGG) database (accessed on 1 March 2021 from https://www.genome.jp/kegg/pathway.html).

### 4.10. Drug Prediction

Potential target drug prediction was performed using the online drug–gene interaction database [111] (DGIdb, accessed on 2 April 2021 from https://dgidb.org, version 4.2.0) based on selected differentially expressed genes (DEGs). The selected DEGs were selected from the DEGs database of MaPac107, PaCaDD159, and PaCaDD165 (expandable samples versus primary samples), with logFC > 3 or < −3, using an adjusted *p*-value < 0.05. The number of potential target drugs commonly shared by MaPac107, PaCaDD159, and PaCaDD165 was shown using an interactive tool for comparing lists with Venn diagrams (Venny, accessed on 4 April 2021 from https://bioinfogp.cnb.csic.es/tools/venny/index2.0.2.html, version 2.0).

### 4.11. IC_50_ Assay

IC_50_ assays were used to evaluate the chemosensitivity of the primary and expandable cells. Gemcitabine and oxaliplatin were diluted to their desired concentrations. In total, 10,000 cells of MaPac107, 10,000 cells of PaCaDD159, and 5000 cells of PaCaDD165 were seeded into 96-well flat plates. After 48 and 72 h of treatment, 3-(4,5-dimethylthiazol-2-yl)-2,5 diphenyltetrazolium bromide (MTT) assays were performed. Then, 20 µL of 5 mg/mL MTT solution was added to each well and incubated for 4 h at 37 °C. MTT formazan was dissolved in DMSO, and absorbance was measured at 560 nm with reference at 670 nm using a SPARK Plate Reader (Tecan V2.3, Männedorf, Switzerland). Experiments were performed in triplicate. IC_50_ values were generated using a fit curve platform [112] (Fit Logistic 4P model) from JMP 15 (SAS Institute Inc., Cary, NC, USA).

### 4.12. Synthesis of Grx1-roGFP3 Redox Sensor and Transfection of Cells

The ratiometric sensor Grx1-roGFP3 was used to analyse the variation in the glutathione redox potential (E_GSH_). Grx1-roGFP3 was kindly donated by Dr. Manfred Frey (Steinbeis-Innovationszentrum Zellkulturtechnik, C/O University of Applied Sciences Mannheim, Germany). In brief, Grx1-roGFP3 was synthesised using the GENWIZ service from Sigma Aldrich and cloned into pHR’SIN-cPPTSEW [113] via the BamHI and XbaI restriction sites. Lentivirus particles were produced as previously described [114]. A stock of pHRSINGrx1-roGFP3 was prepared and stored at −80 °C. Subsequently, 50 µL of lentivirus was added to a T25 flask at 50% confluence of primary and expandable cells and then incubated at 37 °C in a 5% CO_2_ atmosphere for 24 h. The cells were washed with 1× Dulbecco’s phosphate-buffered saline (DPBS, Biozym Scientific, Hessisch Oldendorf, Germany), and fresh Dresden medium was added. This step was repeated after 12 h. Afterwards, the cells were sorted for 80% of positive GFP signal at 4 °C using a BD FACSAria™ IIIu cell sorter (BD Life Sciences, San Jose, CA, USA) and the corresponding BD FACSDiva 8.0.2 software. Vector stability over passages was confirmed by a qualitative evaluation of GFP using the Carl Zeiss Axio Vert. A1 microscope (431030-9040-000; Jena, Germany). No changes in the GFP expression were observed up to three passages after transfection.

### 4.13. Evaluation of E_GSH_ Variation in 2D and 3D Model

The function of Grx1-roGFP3 was verified after sorting. In total, 10,000 cells of MaPac107, 10,000 cells of PaCaDD159, and 5000 cells of PaCaDD165 were seeded into 96-well flat black plates (Corning, Glendale, AZ, USA) and cultured for 24 h at 37 °C in a 5% CO_2_ atmosphere. The program on the SPARK Plate Reader (Tecan V2.3, Männedorf, Switzerland) was run for 15 cycles, with each cycle lasting 15 s. The baseline was obtained by running the program over the first five cycles. The culture medium was then removed and 100 µL of 100 µM H_2_O_2_ (Carl Roth, Karlsruhe, Germany) was added to each well, with measurements taken over five cycles. Next, 100 µL of 1 mM dithiothreitol (DTT; NeoLab Migge, Heidelberg, Germany) was directly added to each well, and measurements were continued for five cycles. The fluorescence intensity of the cells was detected using excitation wavelengths of 395 and 485 nm. Oxidation of the sensor caused an increase in the emission fluorescence at 528 nm when excited at 485 nm and a decrease in emission fluorescence when excited at 395 nm.

E_GSH_ variations were determined by treatment with three different concentrations of gemcitabine and oxaliplatin. In 2D cultures, the same number of cells were seeded into 96-well flat black plates for functional verification. The next day, the culture medium was removed, and 100 µL of chemotherapeutics, H_2_O_2_ (100 µM), and fresh Dresden medium were pipetted. After treatment for 48 h, the plates were analysed using a reader. In 3D cultures, a total of 3000 cells of each cell line were seeded into 96-well U-bottom plates and incubated for three days at 37 °C in a 5% CO_2_ atmosphere. Cells were exposed to chemotherapeutics on the 4th day, and measurements were taken after 48 h of treatment by the reader to detect the fluorescence. Experiments were performed in triplicate.

### 4.14. Statistical Analysis

Statistical analysis was performed using the SPSS software (version 27.0.1; SPSS, Inc., Chicago, IL, USA). Data are expressed as the mean ± SD of three replicate assays. Comparisons between the groups were performed using an independent t-test. Statistical significance was set at *p* < 0.05.

## 5. Conclusions

In summary, we have used a novel approach to establish expandable primary pancreatic cancer cells using a small lentiviral library. We found that expandable primary pancreatic cancer cells conserved some of the characteristics of primary pancreatic cancer cells, while other cell biological functions were impaired. Although they supply sufficient material for drug response prediction assays, their usefulness in a clinical setting remains to be demonstrated.

## Figures and Tables

**Figure 1 ijms-24-13530-f001:**
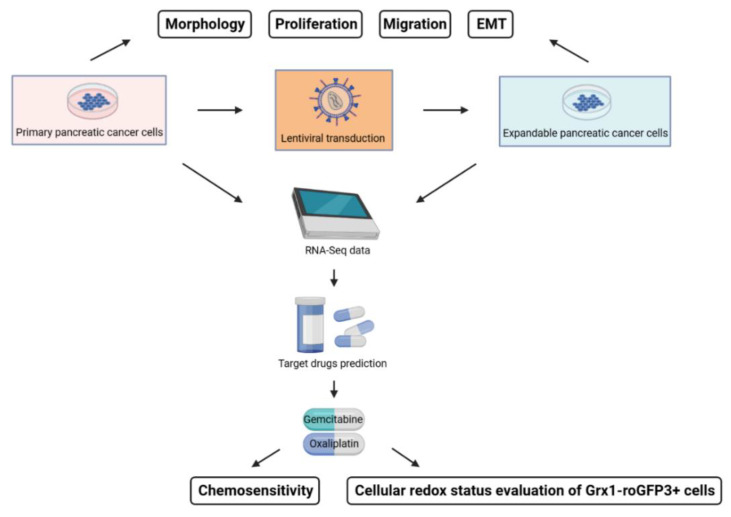
Flowchart of this study.

**Figure 2 ijms-24-13530-f002:**
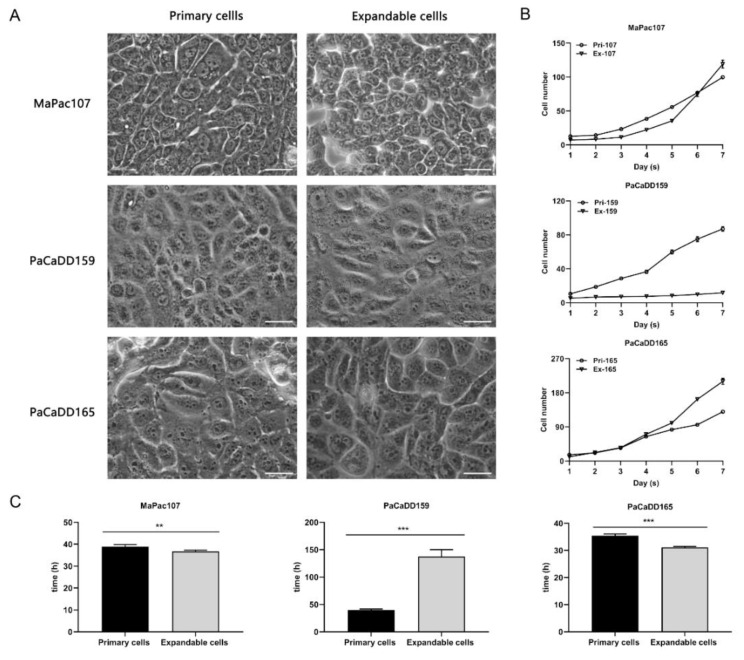
Morphology and growth characteristics of primary and expandable pancreatic cancer cells in 2D culture. (**A**) Morphology of primary and expandable cells. Representative pictures were derived using a Leica DMIRB inverse microscope under 63× magnification. The scale bar was 50 μm. (**B**) Growth kinetic curves of primary and expandable cells. The cell number of each cell line was monitored over 7 days, having the unit for 10 to the power of 4. (**C**) Doubling times were calculated from the logarithmic growth curve according to v = lgN − lgN0/lg2 (t − t_0_), with doubling time = 1/v, where N = number of cells and t = time. ** *p* < 0.01, *** *p* < 0.001.

**Figure 3 ijms-24-13530-f003:**
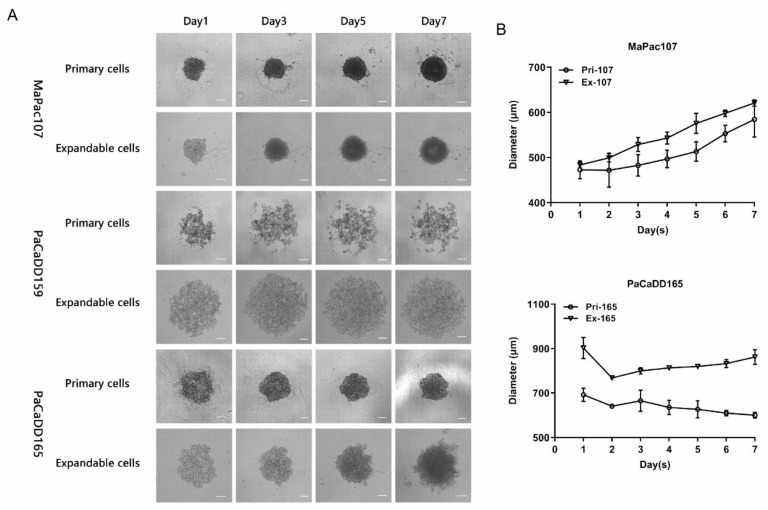
Morphology and growth characteristics of primary and expandable pancreatic cancer cell lines in 3D culture. (**A**) Representative pictures were derived using Carl Zeiss Axio Vert. A1 microscope under 5× magnification. The scale bar was 200 μm. (**B**) The average diameter was measured in the horizontal and vertical axis of spheroid.

**Figure 4 ijms-24-13530-f004:**
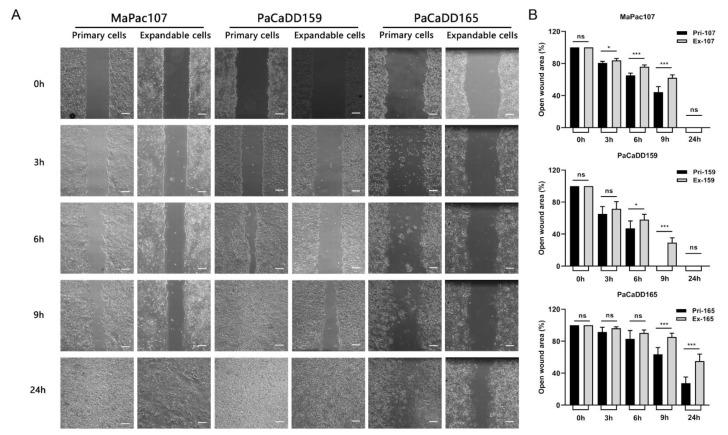
Migration was evaluated in primary and expandable pancreatic cancer cells. (**A**) Time-lapse images at different time points were derived using a Carl Zeiss Axio Vert. A1 microscope under 5× magnification. Scale bar was 200 μm. (**B**) The gap closures were analysed by ImageJ according to wound closure (%) = (original wound area − area at each time point)/original wound area. * *p* < 0.05, *** *p* < 0.001. ns: not significant.

**Figure 5 ijms-24-13530-f005:**
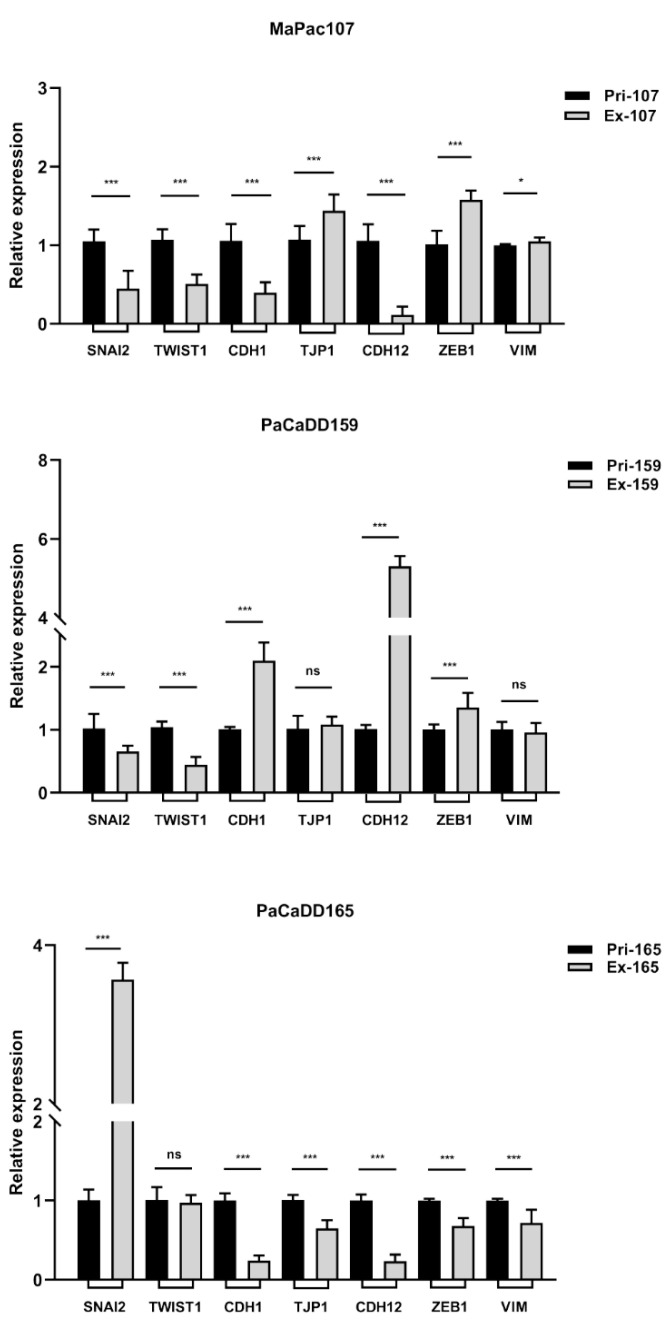
Expression of genes related to EMT was evaluated in primary and expandable pancreatic cancer cell lines by qPCR using SYBR Green I and GAPDH used as internal reference. * *p* < 0.05, *** *p* < 0.001. ns: not significant.

**Figure 6 ijms-24-13530-f006:**
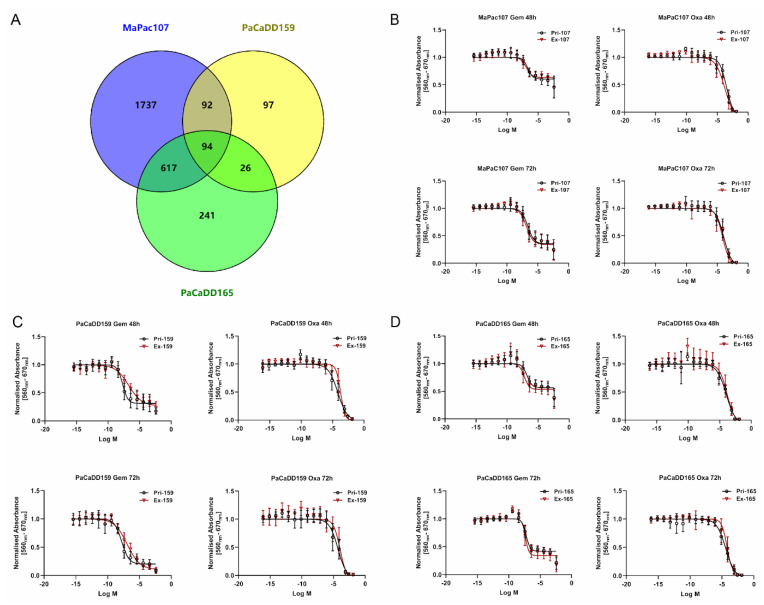
Chemosensitivity of primary and expandable pancreatic cancer cell lines. (**A**) Venn diagram showed the number of potential target drugs commonly shared by MaPac107, PaCaDD159, and PaCaDD165. (**B**−**D**) The dose−response curves of primary and expandable pancreatic cancer cell lines exposed to gemcitabine and oxaliplatin after 48 and 72 h.

**Figure 7 ijms-24-13530-f007:**
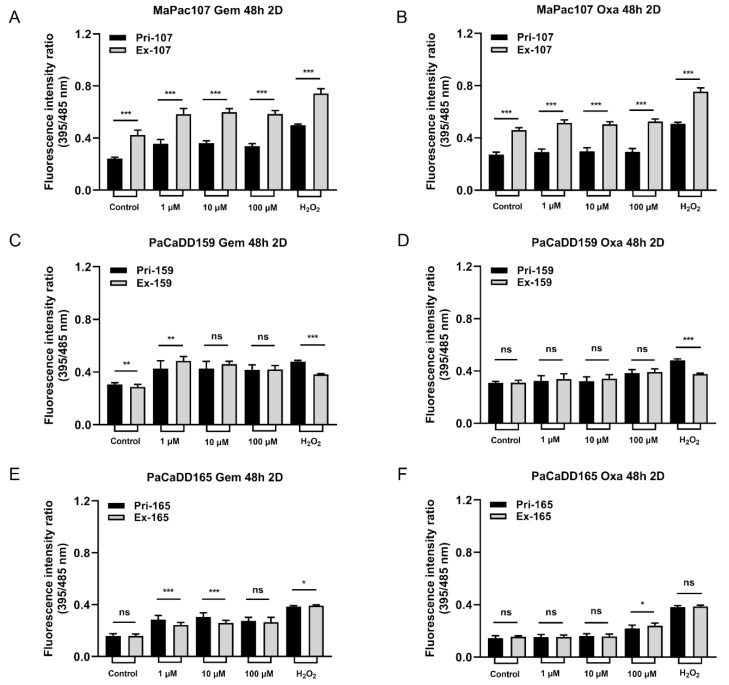
Redox-regulation in response to incubation with gemcitabine and oxaliplatin in primary and expandable pancreatic cancer cell lines expressing Grx1-roGFP3+ in 2D culture. Cells were incubated with three different concentrations of gemcitabine and oxaliplatin and 100 µmol/L H_2_O_2_ for 48 h. Fluorescence intensity was measured by the SPARK Plate Reader. (**A**,**B**) Primary and expandable MaPac107 incubated by compound. (**C**,**D**) Primary and expandable PaCaDD159 incubated by compound. (**E**,**F**) Primary and expandable PaCaDD165 incubated by compound. * *p* < 0.05, ** *p* < 0.01, *** *p* < 0.001. ns: not significant.

**Figure 8 ijms-24-13530-f008:**
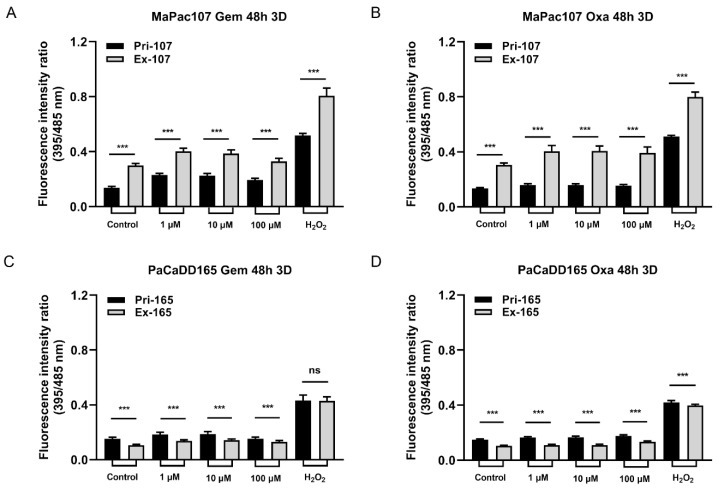
Redox-regulation in response to incubation with gemcitabine and oxaliplatin in primary and expandable pancreatic cancer cell lines expressing Grx1-roGFP3+ in 3D culture. Spheroids were incubated with three different concentrations of gemcitabine and oxaliplatin and 100 µmol/L H_2_O_2_ for 48 h. Fluorescence intensity was measured by the SPARK Plate Reader. (**A**,**B**) Primary and expandable MaPac107 incubated by compound. (**C**,**D**) Primary and expandable PaCaDD165 incubated by compound. *** *p* < 0.001. ns: not significant.

**Table 1 ijms-24-13530-t001:** DEPs related to tumour-specific phenotypes were selected from KEGG database of MaPac107, PaCaDD159, and PaCaDD165 (expandable samples versus primary samples). Each two columns presented normalised enrichment score (NES) and adjusted *p*-values. A. Drug resistance pathways. B. EMT pathways. C. Proliferation pathways. D. ROS-activated pathways. E. Migration pathways.

Pathway Name	MaPac107		PaCaDD159		PaCaDD165	
	NES	Adjusted *p*-Value	NES	Adjusted *p*-Value	NES	Adjusted *p*-Value
A						
NF-kappa B signalling pathway [15]	−1.66	0.0051	2.44	0	−1.42	0.154
EGFR tyrosine kinase inhibitor resistance [16]	−1.51	0.0338	0.93	0.7261	1.07	0.6619
PI3K-Akt signalling pathway [17]	−1.78	0	−1.64	0.0004	−1.15	0.4026
AMPK signalling pathway [18]	−1.63	0.0058	−1.32	0.1306	1.72	0.017
TGF-beta signalling pathway [19]	−1.75	0.0014	−1.15	0.3487	0.96	0.8051
Necroptosis [20]	−1.22	0.1784	1.92	0	0.81	0.9777
Nucleotide excision repair [21]	2.31	0	1.8	0.007	0.96	0.794
Hippo signalling pathway [22]	−1.78	0.0003	−1.01	0.5669	−1.21	0.3943
JAK-STAT signalling pathway [23]	−1.61	0.0065	1.77	0.003	−1	0.7567
MAPK signalling pathway [24]	−1.49	0.0058	−1.02	0.5616	−1.17	0.3943
FoxO signalling pathway [25]	−1.48	0.0211	1.2	0.2548	1.26	0.3474
B						
Toll-like receptor signalling pathway [26]	−1.56	0.0238	1.95	0.0004	−0.54	1
PI3K-Akt signalling pathway [27]	−1.78	0	−1.64	0.0004	−1.15	0.4026
Hedgehog signalling pathway [28]	−1.67	0.0151	1.17	0.3616	0.85	0.8961
JAK-STAT signalling pathway [29]	−1.61	0.0065	1.77	0.003	−1	0.7567
TGF-beta signalling pathway [30]	−1.75	0.0014	−1.15	0.3487	0.96	0.8051
Focal adhesion [31]	−1.89	0	−1.53	0.0122	−1.29	0.2414
Neutrophil extracellular trap formation [32]	−1.27	0.1136	−1.54	0.0245	1.34	0.2414
Hippo signalling pathway [33]	−1.78	0.0003	−1.01	0.5669	−1.21	0.3943
NF-kappa B signalling pathway [34]	−1.66	0.0051	2.44	0	−1.42	0.154
C						
PI3K-Akt signalling pathway [35]	−1.78	0	−1.64	0.0004	−1.15	0.4026
NF-kappa B signalling pathway [36]	−1.66	0.0051	2.44	0	−1.42	0.154
EGFR tyrosine kinase inhibitor resistance [37]	−1.51	0.0338	0.93	0.7261	1.07	0.6619
Hippo signalling pathway [38]	−1.78	0.0003	−1.01	0.5669	−1.21	0.3943
MAPK signalling pathway [39]	−1.49	0.0058	−1.02	0.5616	−1.17	0.3943
Hedgehog signalling pathway [40]	−1.67	0.0151	1.17	0.3616	0.85	0.8961
JAK-STAT signalling pathway [41]	−1.61	0.0065	1.77	0.003	−1	0.7567
Ras signalling pathway [42]	−1.67	0.0005	−1.32	0.1277	−0.94	0.8763
TGF-beta signalling pathway [43]	−1.75	0.0014	−1.15	0.3487	0.96	0.8051
Toll-like receptor signalling pathway [44]	−1.56	0.0238	1.95	0.0004	−0.54	1
Chemokine signalling pathway [45]	−1.75	0.0003	1.91	0.0001	0.92	0.8702
D						
JAK-STAT signalling pathway [46]	−1.61	0.0065	1.77	0.003	−1	0.7567
MAPK signalling pathway [47,48]	−1.49	0.0058	−1.02	0.5616	−1.17	0.3943
PI3K-Akt signalling pathway [49]	−1.78	0	−1.64	0.0004	−1.15	0.4026
NF-kappa B signalling pathway [50]	−1.66	0.0051	2.44	0	−1.42	0.154
Lysosome [51]	−1.65	0.0032	−0.83	0.9192	−1.08	0.6215
AMPK signalling pathway [52]	−1.63	0.0058	−1.32	0.1306	1.72	0.017
FoxO signalling pathway [25]	−1.48	0.0211	1.2	0.2548	1.26	0.3474
E						
EGFR tyrosine kinase inhibitor resistance [53]	−1.51	0.0338	0.93	0.7261	1.07	0.6619
PI3K-Akt signalling pathway [54]	−1.78	0	−1.64	0.0004	−1.15	0.4026
JAK-STAT signalling pathway [55]	−1.61	0.0065	1.77	0.003	−1	0.7567
Toll-like receptor signalling pathway [56]	−1.56	0.0238	1.95	0.0004	−0.54	1
Hippo signalling pathway [57]	−1.78	0.0003	−1.01	0.5669	−1.21	0.3943
Focal adhesion [58]	−1.89	0	−1.53	0.0122	−1.29	0.2414
NF-kappa B signalling pathway [59]	−1.66	0.0051	2.44	0	−1.42	0.154
Hedgehog signalling pathway [60]	−1.67	0.0151	1.17	0.3616	0.85	0.8961
TGF-beta signalling pathway [61]	−1.75	0.0014	−1.15	0.3487	0.96	0.8051
Chemokine signalling pathway [62]	−1.75	0.0003	1.91	0.0001	0.92	0.8702

**Table 2 ijms-24-13530-t002:** Clinical pathological characteristics of the three primary cell lines derived from patients with pancreatic cancer.

Cell Line	Patient	Age (y)	Histology	Grading	Sample Location
MaPac107	Male, Caucasian	77	Ductal Adenocarcinoma	G2	Pleural effusion
PaCaDD159 [102]	Male, Caucasian	78	Ductal Adenocarcinoma	G2	Primary
PaCaDD165 [102]	Male, Caucasian	54	Ductal Adenocarcinoma	G3	Ascites

**Table 3 ijms-24-13530-t003:** Primers used in the study.

Gene Symbol	Qiagen Category Number
GADPHP	QT00273322
SNAI2	QT00044128
TWIST1	QT00011956
CDH1	QT00080143
TJP1	QT00077308
CDH12	QT00018963
ZEB1	QT00020972
VIM	QT00095795

## Data Availability

The data presented in this study are available on request from the corresponding author. The data are not publicly available due to privacy related restrictions.

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
