# Peer review of "Comparison of Tumour-Specific Phenotypes in Human Primary and Expandable Pancreatic Cancer Cell Lines"

_ijms, 2023, doi:10.3390/ijms241713530_

Round 1

Reviewer 1 Report

IJMS (ISSN 1422-0067)

The manuscript titled "Comparison of tumor-specific phenotypes in human primary and expandable pancreatic cancer cell lines" by Guo et al is an intriguing and well-structured study. The study focuses on the establishment and characteristics of expandable cell lines derived from human primary pancreatic cultures. However, in order to achieve their objectives, it is necessary for the authors to conduct additional experimental validations, provide further justifications, and make corrections where needed. These steps are crucial to enhance the accuracy and credibility of their study.

1.     The authors conducted an analysis of the doubling time for three primary cell lines and their expandable counterparts. However, they did not provide a comprehensive description of these results in the relevant section of the paper.

 2.     The graphical representation of the data appears confusing (Figure 2C) in several aspects, when comparing MaPac107 and PaCaDD165 with their expandable cell lines. Although the graph indicates a significant difference between the two, upon closer examination, it does not appear to be statistically significant. Furthermore, there is an inconsistency between the horizontal and vertical representation of the graph. The horizontal axis is labeled as 168 hours; however, the vertical axis is labeled with a different timepoint range of 0-50 h or 0-40 h. This mismatch in the timepoint ranges should be addressed for clarity.

 3.     The authors claim that there was no senescence after five rounds of passage. Within the paper, the authors should provide specific information about the experimental methods utilized to assess senescence and show the corresponding data. It is recommended that techniques such as senescence-associated beta-galactosidase (SA-β-gal) staining, cell cycle analysis, or examination of senescence markers be employed to investigate senescence.

 4.     Authors should modify the sentence for modifying repetition of words (line no. 317-19).

 5.     It is suggested that the labelling of cell line names in Figure 4A should be adjusted to face inward for better readability.

 6.     The authors should consider analyzing selected EMT markers at the protein level to provide a more comprehensive understanding.

 7.     The authors noted variability in results regarding cell proliferation, EMT, and other factors across all three cell lines and their counterparts. It is expected that the authors must provide explanations for any discrepancies in their findings. However, while the discrepancy of PaCaDD159 has been addressed in the discussion, the attenuation of migration ability or therapeutic aspect in all expandable cell lines has not been adequately explained in the discussion section.

 Minor editing of English language is required.

Author Response

Dear Reviewer 1,

Reviewer 2 Report

This is a very novel study comparing primary and expanded cell lines phenotypes PDAC biopsy samples. 

While this is a very interesting study and sheds light on alteration of phenotypes when expanding the cell lines by genome alterations by Lenti virus agent making 30 alterations.

Comments:

1. Were all 30 alterations confirmed, or some failed in different primary lines due to the mutations in the original lines or superseded the original alterations?

2. There are no non-cancerous controls. Why was there no base sample with immortalized biopsy cells from a biopsy tested negative for cancer to negate base changes by the lentiviral immortalization. 

3. Which drug resistance profile (primary vs. expanded) matched the patient response?

4. Why didn't authors tried less aggressive immortalization to preserve primery cell phenotype. A better and more translatable study would have been to test different immortalization protocols capable of preserving primary phenotype. 

Minor corrections, especially with missing articles here and there

Author Response

Dear Reviewer 2,
